# Studying the Interaction of Neutrophils and Glaesserella Parasuis Indicates a Serotype Independent Benefit from Degradation of NETs

**DOI:** 10.3390/pathogens11080880

**Published:** 2022-08-04

**Authors:** Marta C. Bonilla, Simon Lassnig, Andrea Obando Corella, Rabea Imker, Peter Valentin-Weigand, Maren von Köckritz-Blickwede, Anne-Marie Luther, Isabel Hennig-Pauka, Nicole de Buhr

**Affiliations:** 1Department of Biochemistry, University of Veterinary Medicine Hannover, Foundation, 30559 Hannover, Germany; 2Research Center for Emerging Infections and Zoonoses (RIZ), University of Veterinary Medicine Hannover, Foundation, 30559 Hannover, Germany; 3Escuela de Medicina Veterinaria, Universidad Nacional, Heredia 40104, Costa Rica; 4Institute for Microbiology, University of Veterinary Medicine Hannover, Foundation, 30173 Hannover, Germany; 5Unit for Reproductive Medicine, Clinic for Pigs and Small Ruminants, University of Veterinary Medicine Hannover, Foundation, 30559 Hannover, Germany; 6Clinic for Swine, Small Ruminants and Forensic Medicine and Ambulatory Service, University of Veterinary Medicine Hannover, Foundation, 30173 Hannover, Germany; 7Field Station for Epidemiology, University of Veterinary Medicine Hannover, 49456 Bakum, Germany

**Keywords:** neutrophils, neutrophil extracellular traps (NETs), bacterial survival, reactive oxygen species, *Glaesserella parasuis* serotypes

## Abstract

*Glaesserella (G.) parasuis* is one of the most important porcine pathogens causing Glaesser’s disease. Neutrophil granulocytes are the major counteracting cell type of the innate immune system, which contribute to the host defense by phagocytosis or the formation of neutrophil extracellular traps (NETs). Recently, NET-formation has been shown to facilitate the survival of bacteria from the *Pasteurellaceae* family. However, the interaction of NETs and *G. parasuis* is unclear so far. In this study, we investigated the interplay of three *G. parasuis* serotypes with porcine neutrophils. The production of reactive oxygen species by neutrophils after *G. parasuis* infection varied slightly among the serotypes but was generally low and not significantly influenced by the serotypes. Interestingly, we detected that independent of the serotype of *G. parasuis,* NET formation in neutrophils was induced to a small but significant extent. This phenomenon occurred despite the ability of *G. parasuis* to release nucleases, which can degrade NETs. Furthermore, the growth of *Glaesserella* was enhanced by external DNases and degraded NETs. This indicates that *Glaesserella* takes up degraded NET components, supplying them with nicotinamide adenine dinucleotide (NAD), as this benefit was diminished by inhibiting the 5′-nucleotidase, which metabolizes NAD. Our results indicate a serotype-independent interaction of *Glaesserella* with neutrophils by inducing NET-formation and benefiting from DNA degradation.

## 1. Introduction

*Glaesserella (G.) parasuis* is a Gram-negative bacterium of the *Pasteurellaceae* and one of the most important pathogens of the pig, causing Glaesser’s disease, which leads to polyserositis, meningitis and arthritis. This disease causes mortality rates of up to 10% in young pigs and displays a significant economic burden for the pig industry [1]. *G. parasuis* is present in all countries with pig husbandry [2]. Glaesser’s disease is initiated by transmission and colonization of *G. parasuis* in the upper respiratory tract, followed by its proliferation and infection of the lower respiratory tract. There the bacteria disrupt the mucosal barrier allowing them to spread systemically via the blood or lymphatic stream [3]. In healthy pigs, the bacteria are present as a commensal organism in the upper respiratory tract, and they are characterized by a nicotinamide adenine dinucleotide (NAD) dependent growth [4,5]. However, in case of immune imbalance, *G. parasuis* may cause clinical infections. Until today there are 15 different serotypes described, which possess varying levels of virulence. Diagnostic tools based on virulence-associated proteins were developed to identify virulent *G. parasuis* strains. Autogenous vaccines are produced for the respective strain responsible for disease in a herd because no commercial cross-protective vaccine is available. These vaccines have the potential to decrease the mortality rate, but due to the fact that they provide only poor cross-protection, vaccine failures are common, as reviewed by Macedo et al. [6].

After the spread into the lower respiratory tract, *G. parasuis* encounters cells of the innate immune system, such as alveolar macrophages and neutrophil granulocytes. Especially virulent strains can avoid phagocytosis by porcine alveolar macrophages [7]. A similar attribution can be seen regarding the complement system, which fails to kill virulent strains [8]. This is one explanation why only strains classified as virulent are found in the lungs of infected pigs [4]. The cell wall of *G. parasuis* contains lipopolysaccharide molecules that can activate the NF-κB pathway via toll-like receptors (TLR) and cytokine production, including IL-8 [9]. IL-8 is known to attract neutrophils chemotactically to the site of infection [10]. It has been shown in vitro and in vivo that neutrophils can phagocytose and kill *G. parasuis* [11].

Neutrophils are one of the most abundant leukocytes in pigs, with numbers up to 70% of the white blood cells, and display the first line of defense against invading pathogens, as they are recruited to the site of infection as the first cell type [12,13]. They possess several effector functions to fight intra- and extracellularly against invading pathogens. During phagocytosis, microbes become engulfed and digested intracellularly. The degranulation of vesicles from the cytoplasm releases antimicrobial peptides for extracellular killing. In addition to this, the production of reactive oxygen species (ROS) displays another potent defense mechanism. ROS can damage microbes intracellularly [14].

The formation of neutrophil extracellular traps (NETs) is an extracellular mechanism of neutrophils to partially kill and prevent the further spread of pathogens in the tissue [15]. Upon activation by cytokines such as IL-8 or TNF-α or by direct pathogen contact, the neutrophil decondenses its nuclear DNA in a ROS-dependent process after NADPH oxidase activation called NETosis [16]. The DNA fibers are decorated with histones and antimicrobial molecules and are released by a rupture of the cellular membrane. Furthermore, ROS-independent NET formation is described. The induction of NET formation has been shown by all types of pathogens [14,17,18,19].

For *Actinobacillus pleuropneumoniae* (*A.pp*), which is also a member of the *Pasteurellaceae,* it was shown by us that degraded NETs cause a growth enhancement [20]. Degraded NETs release NAD and adenosine, which promotes the growth of *A.pp*. This phenotype could be abolished by inhibiting the 5′-nucleotidase. This is described as an essential enzyme in the NAD and adenosine metabolism of cells and bacteria [21]. *A.pp,* as a NAD-dependent growing bacteria, benefits from the host defense mechanism. Next to *A.pp*, we showed a similar phenotype for the human pathogen *Haemophilus influenzae,* also a *Pasteurellaceae* member, in the presence of degraded human NETs, which also benefits from the host defense mechanism [20]. As *G. parasuis* also grows NAD dependently, similar phenotypes are conceivable. Therefore, we investigated in this study the interaction of three different *G. parasuis* strains with porcine neutrophils. We used *G. parasuis* strains of serotypes 7, 13 and 15. Apathogenic serotype 7 strains were described [22], but this serotype can also be isolated from systemic infections [23]. The *G. parasuis* serotype 7 strain used in this study was isolated from a pig with meningitis. The serotype 13 strain used here was isolated from a pig with pneumonia. Serotype 13 strains are often considered to be highly virulent [23]. Serotype 15 strains were described as moderately virulent [24]. The serotype 15 strain used in this study was isolated from a diseased pig from an unknown organ.

Firstly, these three strains were chosen to compare isolates from different origin organs. Secondly, these serotypes cover the range of virulence known for *G. parasuis,* from avirulent to highly virulent. With serotypes 7 and 13, we included two of the most common serotypes in North America, Europe and Asia [25].

We investigated the host side different effector functions of the neutrophils to clear an infection, namely NET formation, ROS production and the overall killing behavior of neutrophils. Furthermore, on the pathogen side, we investigated the growth, the DNase activity and the survival in the presence of degraded NETs.

## 2. Results

### 2.1. G. parasuis Serotypes Show a Tendency to Induce ROS Production

The production of ROS is a key effector function of neutrophil granulocytes, plays a role in the NET formation and is needed to digest phagocytosed pathogens. The incubation with *G. parasuis* for 1 h stimulated porcine neutrophils to produce slightly elevated amounts of ROS compared to the medium control measured by flow cytometry (Figure 1). However, a remarkable difference from the untreated control sample was detectable (*p* = 0.06) only after incubation with *G. parasuis* ST 7. For *G. parasuis* ST 13 and 15, the ROS production was higher than the negative control, but no significant difference was detected. In summary, *G. parasuis* only slightly induces an oxidative burst of neutrophils after 1 h stimulation.

### 2.2. G. parasuis Stimulates Neutrophils to Release the Maximum Amount of NET Activated Cells within One Hour of Infection

In order to analyze if the different serotypes induce NETs in comparable amounts, we performed NET induction assays with all three serotypes grown to a similar growth phase, adjusted them all to a multiplicity of infection (MOI) of two and performed a time kinetic analysis. After one hour of infection with the three different serotypes of *G. parasuis*, all of them significantly induced NET-activated cells. Interestingly, after three hours of infection, the percentage of NET-activated cells only further increased in the positive control but not after *G. parasuis* infection (Figure 2). This indicates that the highest NET release by *G. parasuis* stimulation occurs within the first hour. Thus, it may be speculated that a conceivable DNase activity of *G. parasuis* could have an influence on the outcome.

### 2.3. BLAST Analysis Revealed No Hint for DNase(s) in G. parasuis. However, Dnase Activity Was Detected in Supernatants of G. parasuis Serotypes by a Sensitive DNase Activity Test

Other bacteria are known to produce DNases to degrade NETs and thereby escape from NETs (Table 1). As the NET-activated cells were low three hours after *G. parasuis* infection, we analyzed whether a DNase of *G. parasuis* could explain this phenotype. However, BLAST-analysis of described and characterized NET degrading, secreted nucleases of bacteria in the genome sequence of four *G. parasuis* strains did not give any hint for the presence of secreted nucleases in the *G. parasuis* genome (Table 1). Nevertheless, we analyzed the supernatants of the three different *G. parasuis* serotypes included in this study for DNase activity with two different methods. Firstly, eukaryotic DNA was incubated in the presence of a DNase activity buffer that is needed for the activity of, for example, *S. suis* DNases SsnA and EndAsuis [26,27]. This buffer contains a final concentration of 1.5 mM MgCl_2_ and 1.5 MM CaCl_2_ and is adjusted to a pH of 7.4. However, under the chosen conditions, after an incubation of 24 h, no DNase activity was detectable (Figure 3A). Therefore, we analyzed all samples with a more sensitive DNase activity assay to determine the enzymatic activity in pmoL/min/mL. Interestingly, we could detect DNase activity in all serotypes with the highest activity in the overnight cultures. Serotypes 7 and 13 showed DNase activity twice as strong compared to serotype 15 in the overnight culture (Figure 3B–D). An increase in DNase activity was identified over time in all serotypes. The overnight cultures showed comparable DNase activity for serotypes 7 and 13 with a DNase activity around 31 pmoL/min/mL, whereas in serotype 15, the DNase activity of the overnight culture was measured with only 13.5 pmoL/min/mL. Further studies are needed to clarify under which conditions *G. parasuis* shows DNase activity, if a difference between serotypes exists and if the DNase can indeed degrade NETs.

### 2.4. External DNase, Neutrophils and Degraded NETs Enhance Growth of Different G. parasuis Serotypes

After we identified that *G. parasuis* induces NETs, we investigated in the next step if *G. parasuis* is killed in the presence of NETs or survives, such as *A.pp* in the presence of degraded NETs. The survival assay with the three serotypes of *G. parasuis* was conducted with MOI = 1 and showed that all of them can obtain benefits from DNase, neutrophils and degraded NETs. Interestingly, *G. parasuis* ST 7 reaches the highest survival factor (SF) in the presence of degraded NETs (ST 7: mean SF = 24.7) compared to the other two serotypes (ST 13: mean SF = 5.8; ST 15: mean SF = 10.5) (Figure 4A–C). Whereas ST 7 and ST 15 were at all not efficiently eliminated by neutrophils (ST 7 SF = 12.47; ST 15 SF = 3.45), ST 13 was efficiently killed by neutrophils (ST 13 SF = 0.52), showing a serotype-dependent survival of *G. parasuis*. These phenotypes show how different all three serotypes are influenced by neutrophils and degraded NETs. However, all three serotypes clearly benefit from degraded NETs and/or degraded DNA of dying bacteria.

### 2.5. Enhancement of the Growth of G. parasuis Serotype 15 Dependents on the 5′-Nucleotidase

In order to analyze whether the identified phenotype of *G. parasuis* in the presence of degraded NETs depends on the 5′-nucleotidase, we conducted assays in the presence of a 5′-nucleotidase inhibitor. Neutrophils were incubated with fresh grown and washed *G. parasuis* ST 15 with an MOI of 0.5. Under the chosen conditions, this serotype grows in the presence of degraded NETs (Figure 5). As expected, the growth was significantly altered in the presence of degraded NETs and a coincubation with a 5′-nucleotidase inhibitor that blocks the enzymatical metabolism of adenosine and NAD.

## 3. Discussion

As *A.pp* can strongly induce NETs and benefits from free available components of degraded NETs [20], we investigated whether *G. parasuis*, belonging as well to the family of the *Pasteurellaceae*, shows similar phenotypes.

Neutrophils release NETs as a response to pathogens and cytokines [16]. *G. parasuis* is statistically significant in inducing NETs compared to the medium control (Figure 2). However, compared to *A.pp*, which induces in porcine neutrophils up to 70 % NET activated cells after three hours of infection, *G. parasuis* induces fewer NET activated cells in porcine neutrophils (10–15%, Figure 2). The number of NETs do not differ vastly between the three serotypes. The major amount of NET formation occurs during the first hour of infection (Figure 2B), as further incubation did not increase the number of NETs (Figure 2C). This could indicate that the pathogenicity of a serotype does not influence the neutrophil reaction regarding NETosis.

Furthermore, in the case of *A.pp*, an increase in NET-activated cells was observed after infection of neutrophils over time [20], which was not detectable in the case of *G. parasuis* (Figure 1). Two reasons are plausible: Firstly, in *G. parasuis,* no toxins are described that could be a potential NET inducer, such as *A.pp* producing repeats in toXin (RTX)-toxins that attack neutrophils and/or, secondly, *G. parasuis* produces a NET degrading DNase, such as *S. suis* [27] that degrades NETs, and therefore a detection by immunofluorescence microscopy is influenced. Regarding the former explanation, there is no comparable toxin like the RTX-toxins described in *G. parasuis*. Therefore, this would explain the low amount of NET releasing neutrophils after *G. parasuis* infection. However, a combination with the latter reason is also possible. Nevertheless, we did not identify with BLAST analysis (Table 1) and a classical DNase activity test (Figure 3A) any hint for a DNase that is described in other bacteria as NET degrading DNase in the supernatant of *G. parasuis,* such as in previous studies for example for *S. suis* [26,27]. One possible candidate for a NET degrading DNase of *G. parasuis* is a cytolethal distending toxin (Cdt), as a DNase activity was described for the purified complex CdtB [44]. A homologous CdtB in *Haemophilus ducreyi* was detected in the culture supernatants [45]. This conforms with our findings of detected DNase activity in the supernatants of *G. parasuis* with a more sensitive assay (Figure 3B). Further studies are needed to investigate if CdtB is a NET degrading DNase of *G. parasuis*. A promising approach would be NET degradation assays with *cdtB* deletion mutants compared to the wild type or the incubation of NETs with purified CdtB. Furthermore, if a NET degrading DNase activity is serotype dependent and thereby influencing the pathogenicity needs further investigations. However, the comparison of DNase activity between the three serotypes shows differences and leads to the hypothesis that DNase activity capacity could influence the host-pathogen interaction.

Another effector function of neutrophils that we investigated in our study is the production of ROS. The production of ROS represents one of the key effector mechanisms of neutrophils against microbial infections. Increasing ROS levels are an indicator of neutrophil activation [46]. The major part of neutrophil ROS production is performed via the NADPH oxidase (NOX2) being triggered by immune stimuli or by phagocytosis, where cellular oxygen is reduced to several ROS types [47,48]. Interestingly, the pathogenic human relative of *G. parasuis*, *Haemophilus influenzae*, has the OxyR system as a counter mechanism against ROS. It detects ROS and expresses numerous defense antioxidants fighting against respiratory burst, making it a crucial survival tool in in vivo infections [49]. In *G. parasuis,* OxyR is described as a transcription factor that is important for oxidative stress resistance [50]. The incubation of three serotypes of *G. parasuis* with freshly isolated porcine neutrophils showed only a slight increase in ROS levels compared to the medium control (Figure 1). As NET formation is partially ROS dependent [46], at least a ROS-dependent NET release by *G. parasuis* with high amounts is not expected. However, a ROS-independent NET release is possible, and further studies should investigate the NET formation mechanisms by *G. parasuis* in more detail.

As we investigated the NET formation and ROS production by neutrophils in response to *G. parasuis* and its DNase activity and growth, we were finally interested in whether the survival of *G. parasuis* serotypes in the presence of neutrophils and degraded NETs differs between the serotypes.

All three serotypes showed a significant growth increase after incubation with the DNase mix (Figure 4), as was observed for *A.pp* in our previous study [20]. We assumed that the bacteria could use the DNA of dying bacteria. The survival is further increased in serotypes 7 and 15 in the presence of degraded NETs (neutrophils + DNase mix). This again shows a similar phenotype as it was observed with *A.pp* [20].

The data in Figure 5 show for serotype 15 that *G. parasuis* can take up and benefit from degraded NETs, as the enhanced growth was abolished by incubation with a 5′-nucleotidase inhibitor. This enzyme hydrolyses nucleotides as adenosine monophosphate to adenosine, being a part of the group of enzymes involved in the bacterial NAD metabolism [21]. This indicates that the growth effect is directly linked to the bacterial NAD metabolism, similar to our findings in *A.pp* [20].

In conclusion, we identified that serotypes 7, 13 and 15 of *G. parasuis* induce only slightly ROS production, which indicates that the observed NET induction is ROS-independent. NET-activated cells can be observed, although all *G. parasuis* serotypes release a DNase. Therefore, *G. parasuis* can escape from NETs, and all investigated serotypes benefit in their growth from degraded NETs. As *G. parasuis* produces its own DNase, another external DNase source, as described for *A.pp*, this is not obligatory needed for enhanced growth. However, the external DNase is further supporting the growth of *G. parasuis*. If a pig is infected, other host-defense mechanisms of neutrophils than NET formation are needed to counteract a severe infection, such as for example, antibody-based phagocytosis. The interaction of neutrophils and *G. parasuis* serotypes show partial differences and varying degrees of intensity that should be investigated in future studies in more detail. These differences may explain why some strains can lead to meningitis and others cannot.

## 4. Materials and Methods

### 4.1. PPLO Medium

Fresh self-made PPLO media were made for the culture of *G*. *parasuis*. In 1 L distilled water 21 g of BD Difco™ Dehydrated Culture Media: Mycoplasma Broth (PPLO Broth without Crystal Violet) (BD 255420, Heidelberg, Germany) was added. After autoclaving, the medium was slightly cooled down, and 10 mL of a 10% solution of Tween^®^ 80 (9139.1, Carl Roth^®®^, Karlsruhe, Germany) diluted in distilled water was added. In addition, as NAD enrichment, 10 mL of IsovitaleX was added. The substituted IsovitaleX was prepared as follows: to 100 mL distilled water, 0.1 g of L-Cystine-dihydrochloride (C6727-25G, Sigma Aldrich, Munich, Germany) was added and the pH adjusted to 9.4 with 5 M NaOH. Then, 2.6 g of L-Cysteine hydrochloride (W778567-100G, Sigma Aldrich, Munich, Germany) was added and properly dissolved. Afterward, 5 g of D(+)-Glucose Monohydrate (6887.1, Carl Roth^®®^, Karlsruhe, Germany) was added, and the pH was adjusted between 6 and 7 with 5 M NaOH. In the end, 0.1 g of NAD, free acid, grade II (REF10127981001, Roche, Mannheim, Germany) was added, and the solution was sterile filtrated. Aliquots from the IsovitaleX were stored at −20 °C until use.

### 4.2. Glaesserella (G.) parasuis Strains and Growth Conditions

Three different *G. parasuis* strains were included in this study: *G. parasuis* serotype 7, isolated from a diagnostic sample from a diseased pig with meningitis (brain; lab number 21/803); *G. parasuis* serotype 13, isolated from a diagnostic sample from a diseased pig with pneumonia (lung; lab number 21/981); and *G. parasuis* serotype 15, isolated from a diagnostic sample from a diseased pig (unknown organ; lab number: 18/001).

*G. parasuis* was grown for experiments in Figure 5 as follows: on boiled-blood plates with NAD (257011- BD™ Chocolate Agar, Blood Agar No. 2 Base. Heidelberg, Germany) at 37 °C and 5% CO_2_. The liquid culture was grown in a fresh PPLO medium. An amount of 10 mL PPLO media was inoculated with one colony-forming unit from the boiled-blood plate (weekly new spread out from −80 °C cryostock) and incubated for 16 h in a culture tube at 37 °C and 5% CO_2_. On the next day, 5 mL of over-night-culture were transferred to 45 mL of pre-warmed PPLO medium and afterward incubated in a rotation shaker (200 rpm, 37 °C) until reaching the OD_600nm_ = 0.35 to 0.4 (mid-log-phase). Then, 2 mL bacteria suspension was centrifuged (2600× *g*, 5 min), and the pellet of bacteria was twice washed with 1× PBS. The bacteria suspension was adjusted in PBS to an OD_600nm_ = 0.55 to 0.6. A diluted sample was added to the survival assays, and finally, an MOI of 0.5 was reached.

#### Cryostocks for NET Induction and Survival Assay

A streak out from *G. parasuis* ST 7, *G. parasuis* ST 13 and *G. parasuis* ST 15 was made on chocolate agar plates (257011- BD™ Chocolate Agar, Blood Agar No. 2 Base. Heidelberg, Germany) and incubated for 20 h (37 °C, 5% CO_2_). An overnight culture with three colonies in 14 mL Simport tubes (Culture tubes polypropylene graduated, EC04.1, Carl Roth^®®^, Karlsruhe, Germany) with 10 mL of supplemented PPLO medium for each strain was prepared. *G. parasuis* ST 13 and ST 15 were incubated for around 16 h (37 °C, 5% CO_2_) in a melting ice bath to delay the start of growth and were then incubated for 1 h in the shaking incubator (200 rpm, 37 °C). *G. parasuis* ST 7 was incubated in the shaking incubator (200 rpm, 37 °C, 0% CO_2_) for around 16 h. A 1:10 dilution from all overnight cultures was made in a pre-warmed PPLO supplemented medium (total 50 mL) and incubated in the shaking incubator (200 rpm, 37 °C, 0% CO_2_) until it reached the late exponential growth phase with an optical density (OD_600nm_) of 0.48 ± 0.01(*G. parasuis* ST 13), 0.40 ± 0.01 (*G. parasuis* ST 7) and 0.52 ± 0.01 (*G. parasuis* ST 15). The bacterial suspension was mixed with glycerol (final concentration 15%). Aliquots were immediately frozen in liquid nitrogen and stored at −80 °C until usage. Each cryostock was used only once.

### 4.3. BLAST Analysis

The BLAST analysis was conducted with NCBI Blastp, including a search that was limited to: *G. parasuis* (taxid: 738), *G. parasuis* 29755 (taxid: 456298), *G. parasuis* D74 (taxid: 1275971) and *G. parasuis* SH0165 (taxid: 557723).

### 4.4. DNase Activity Assays

Supernatants from overnight cultures and growth curves of all *G. parasuis* serotypes were collected after centrifugation (3000× *g*, 5 min). They were stored at −80 °C and analyzed with two different assays. 1. Bacterial supernatants were mixed with 50 µL DNase buffer (3 mM MgCl_2_, 3 mM CaCl_2_, 300 mM Tris, pH 7.4) in a ratio of 1:2. The final volume of the sample was 100 µL. Calf thymus DNA (0.5 µg) was then added. The samples were incubated at 37 °C. After 24 h incubation, a visual examination of DNA for DNase activity evaluation was conducted after 1% agarose gel electrophoresis and staining of DNA with ROTI®GelStain Ready-To-Use (3865.1, Roth, Karlsruhe, Germany). 2. DNase 1 assay Kit (ab234056, Abcam, Discovery Drive, Cambridge Biomedical Campus, Cambridge, CB2 0AX, UK) was used to determine the DNase activity in the supernatant. The test was performed following the manufacturer’s instructions with 25 µL for each sample. PPLO medium Dnase activity was measured, calculated and substracted from the supernatant results.

### 4.5. Purification of Porcine Neutrophils

Fresh heparinized blood was collected from healthy pigs in S-Monovette^®^ Lithium-Heparin 9 mL tubes (Sarstedt, Nümbrecht, Germany). The donor pigs were kept either in the Clinic for Swine, Small Ruminants and Forensic Medicine and Ambulatory Service at the University of Veterinary Medicine Hannover, Foundation, Germany, for sperm donation or in the Research Center for Emerging Infections and Zoonoses from the University of Veterinary Medicine Hannover, Foundation, Germany, as control animals. Pigs included in this study were male and female and were not under treatment, and a general health examination by a trained veterinarian was conducted before the blood donation. The blood was used immediately for neutrophil isolation. Porcine neutrophils were purified using Biocoll (1.077 g/mL, Biochrom, L6115, Berlin, Germany or Biocoll^®^ (1.077 g/mL, Bio&SELL GmbH, BSL6115, Nürnberg, Germany) and hypotonic lysis of erythrocytes as previously described [51]. Cells were resuspended in Roswell Park Memorial Institute (RPMI) 1640 without phenol red (Thermo Fisher, 11835063 GibcoTM, Waltham, MA, USA).

Within the assays that show data with the three serotypes, always neutrophils from the same donor pig (for one independent experiment) were used for all serotypes in parallel.

### 4.6. NET Induction G. parasuis

Freshly isolated porcine neutrophils (2 × 10^5^ cells per well) were seeded in 48-well plates (Greiner Bio-One, 677102, Kremsmünster, Austria) on cover slides (8 mm; MENZCB00080RA120, VWR International GmbH, Darmstadt, Germany) previously coated with poly-L-lysine (0.01% solution P4707, Sigma Aldrich, Munich, Germany). The neutrophils were infected with *G. parasuis* from cryostocks prepared as described above. The bacterial suspension was added to infect neutrophils from cryostocks with an MOI of 2. As a negative control, Roswell Park Memorial Institute (RPMI) 1640 Medium, without phenol red (Thermo Fisher, 11835063 GibcoTM Waltham, MA, USA) and as a positive control Methyl- β-cyclodextrin (CD) (10 mM final concentration, C4555 Sigma Aldrich, Munich, Germany) was added to the neutrophils in a volume of 200 µL. The plate was centrifuged (250 g, 5 min) and incubated at 37 °C with 5% CO_2_. After 1 or 3 h of incubation, samples were fixed with 4 % paraformaldehyde (final concentration).

### 4.7. Visualization and Quantification of NETs

NETs were stained as previously described [51]. Briefly, after blocking and permeabilization, neutrophils were incubated with a mouse monoclonal antibody (IgG2a) against DNA/Histon 1 (MAB3864; 2.2 mg/mL diluted 1:2000 Billerica, MA, USA) and a polyclonal rabbit anti-human myeloperoxidase antibody (A039829-2 Agilent, Santa Clara, CA, USA, 3.3 mg, 1:309) for 1 h at room temperature. As secondary antibodies, a goat anti-mouse Alexa 488Plus (1:500, A32723, 2 mg/mL, Invitrogen, Carlsbad, CA, USA) and goat anti-rabbit antibody (Alexa633, A21070, 2 mg, Thermo Scientific, Waltham, MA, USA; diluted 1:500 in blocking buffer) were used. For the isotype controls, a murine IgG2a (from murine myeloma, M5409-1mg, concentration 0.2 mg/mL, 1:364 Sigma Aldrich, Munich, Germany) and a rabbit IgG (from rabbit serum, Sigma Aldrich, Munich, Germany, I5006, 1.16 mg, 1:108.75) as first antibodies were used. As second antibodies, those previously described in the other samples were used.

Samples were recorded using a Leica TCS SP5 AOBS confocal inverted-base fluorescence microscope with an HCX PL APO 40× 0.75–1.25 oil immersion objective. Settings were adjusted with control preparations using isotype control antibodies. For each sample, six randomly selected images per independent experiment were acquired and used for quantification of NET-producing cells (NET induction). The cells were counted using ImageJ software (version 1.52q, National Institute of Health, Bethesda, MD, USA). Neutrophils with an evident off-shoot of DNA were counted as positives or if at least two of the following criteria were found: enlarged nucleus, decondensed nucleus or blurry rim. The percentage of NET-positive neutrophils was calculated. An average of six pictures of each sample was made.

### 4.8. G. parasuis ST 15 Survival Assay 

In order to analyze the antimicrobial activity of NETs against *G. parasuis* ST 15, porcine neutrophils and *G. parasuis* were co-incubated. *G. parasuis* (MOI = 0.5, freshly grown and washed) was prepared as described above and incubated with 2 × 10^5^ cells in a total volume of 200 µL (48-well plate). As a growth control, *G. parasuis* ST 15 was grown in RPMI alone. Furthermore, a DNase mix with micrococcal nuclease from *S. aureus* and Deoxyribonuclease I from bovine pancreas was added in the same final concentrations as described previously [20]. The plate was centrifuged (370 g, 5 min) and incubated at 37 °C with 5% CO_2_. In order to determine surviving bacteria, samples were mixed, and serial dilutions were made at time points 0 min and 180 min. Dilutions were plated in duplicates on dried boiled-blood agar plates with NAD and incubated overnight at 37 °C, 5% CO_2_. Colony-forming units were counted, and the survival factor was calculated as described previously [20]. In addition, samples were co-incubated with adenosine 5′-(α,β-methylene) diphosphate (Sigma Aldrich, final concentration 500 µM) to reduce adenosine production by blocking adenosine synthase as previously described [52].

### 4.9. G. parasuis Survival Assays

In order to analyze the antimicrobial activity of neutrophils against *G. parasuis*, porcine neutrophils and *G. parasuis* were co-incubated. *G. parasuis* (MOI = 1) was prepared as described above, and cryostocks were used. The 2 × 10^5^ cells in a total volume of 200 µL (48-well plate, Greiner Bio-One, 677102, Kremsmünster, Austria) were seeded. As growth control, *G. parasuis* was grown in RPMI alone. Furthermore, a DNase mix with micrococcal nuclease from *Staphylococcus aureus* (100 mU final, N5386, Sigma Aldrich, Munich, Germany) and Deoxyribonuclease I from bovine pancreas (DNase I) (20 U final, 18535.02, SERVA Electrophoresis GmbH, Heidelberg, Germany) was added. The plate was centrifuged (300× *g*, 5 min) and incubated for 3 h at 37 °C with 5% CO_2_.

In order to determine the surviving bacteria, samples were mixed, and serial dilutions were made at time points 0 h and 3 h. The different dilutions were plated on chocolate agar plates (257011- BD™ Chocolate Agar, Blood Agar No. 2 Base. Heidelberg, Germany) and incubated for 20 h (37 °C, 5% CO_2_). Colony forming units (CFU/mL) were determined, and the survival factor (SF) was calculated with the formula SF_3h_ = CFU_3h_/CFU_0h_ as described previously [20,53].

In order to analyze the effect of plasma on neutrophil antimicrobial activity against *G. parasuis*, porcine neutrophils and *G. parasuis* were co-incubated in the presence or absence of 20% of autologous plasma. The CFU/mL and the survival factor (SF) were analyzed as described above.

### 4.10. Measurement of ROS

Intracellular ROS production was measured as described previously with small modifications [54]. Briefly, isolated neutrophils were incubated with MOI = 2 of *G. parasuis* of three serotypes 7, 13 and 15 from cryostocks. Unstimulated cells served as a negative control. Tubes were incubated for 30 min at 37 °C and 5% CO_2_. Subsequently, 2′,7′-dichlorodihydrofluorescin-diacetate (DCFH-DA; D6883-50MG; Sigma Aldrich, Munich, Germany) with a final concentration of 10 μM was added to each sample, and all samples were incubated at 37 °C and 5% CO_2_ for another 30 min. Samples from three different pigs were analyzed, and a respective background control without DCFH-DA was included. Intracellular ROS production was measured by flow cytometry (Attune^®^NxT Acoustic Focusing Flow Cytometer, Invitrogen; Laser 488 nm (50 mW), filter BL1 = 530/30). The mean green fluorescence intensity of all cells (X-Mean of BL-1) was determined as a relative measurement of ROS production. The gating strategy included only singlets of the neutrophil population (See Figure 5). Data were analyzed with FlowJoTM10.7.1 software (Ashland, OR, USA).

### 4.11. Statistical Analysis

Data were analyzed using Excel 2019 and Excel 365 (Microsoft), and GraphPad Prism 9.1.0 and 9.3.1 (GraphPad Software, San Diego, CA, USA). Data are presented with mean ± SD or mean ± SEM as described in the figure legends. The differences between groups were analyzed as described in the figure legends (* *p* < 0.05, ** *p* < 0.01, *** *p* < 0.001).

## Figures and Tables

**Figure 1 pathogens-11-00880-f001:**
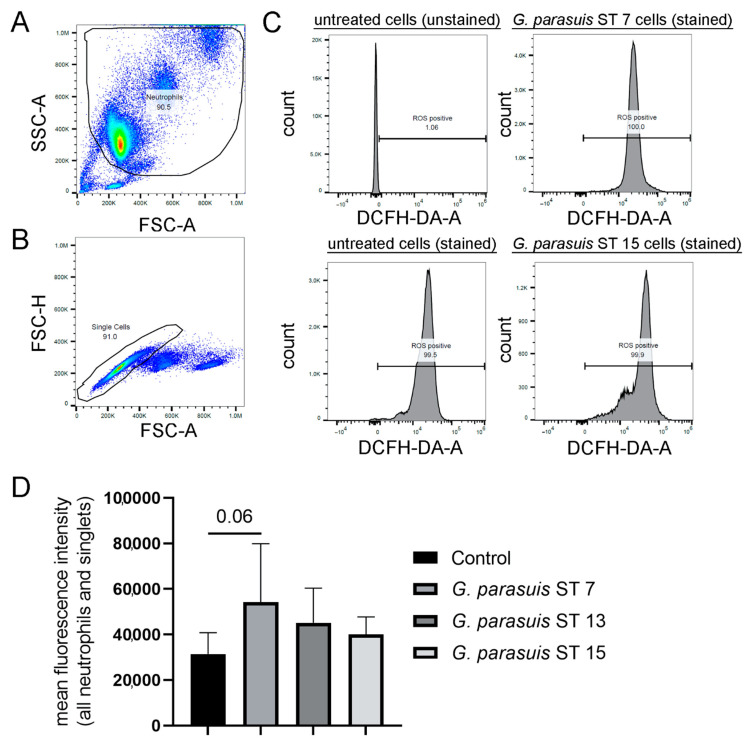
Isolated porcine neutrophils produced slightly elevated amounts of intracellular ROS upon contact with different strains of *G. parasuis*. (**A**–**C**) The intracellular ROS production was determined by adding 2′7′ dichlorodihydrofluorescin-diacetate (DCFH-DA) to unstimulated and *G. parasuis* stimulated cells. The gating strategy for the DCF-positive cells (oxidation of DCFH-DA by ROS results in fluorescence of 2′-7′-dichlorofluorescein) by flow cytometry is presented. (**A**) Based on FSC-A and SSC-A, the neutrophil population was gated. (**B**) Based on FSC-A and FSC-H, all single cells were gated from the neutrophil population. (**C**) The population of ROS-positive cells was gated according to the unstained control, as also unstimulated neutrophils produce ROS. The shift of the peak indicates an alteration of ROS production. Example histograms are presented. (**D**) The mean fluorescence intensity is presented, and the highest fluorescence intensity was determined after *G. parasuis* ST 7 incubation. The untreated and stained cells were used as a control to determine a change in ROS production during the infection with *G. parasuis*. Data were analyzed with one-tailed paired Student’s *t*-test and are presented with mean ± SD (*n* = 3).

**Figure 2 pathogens-11-00880-f002:**
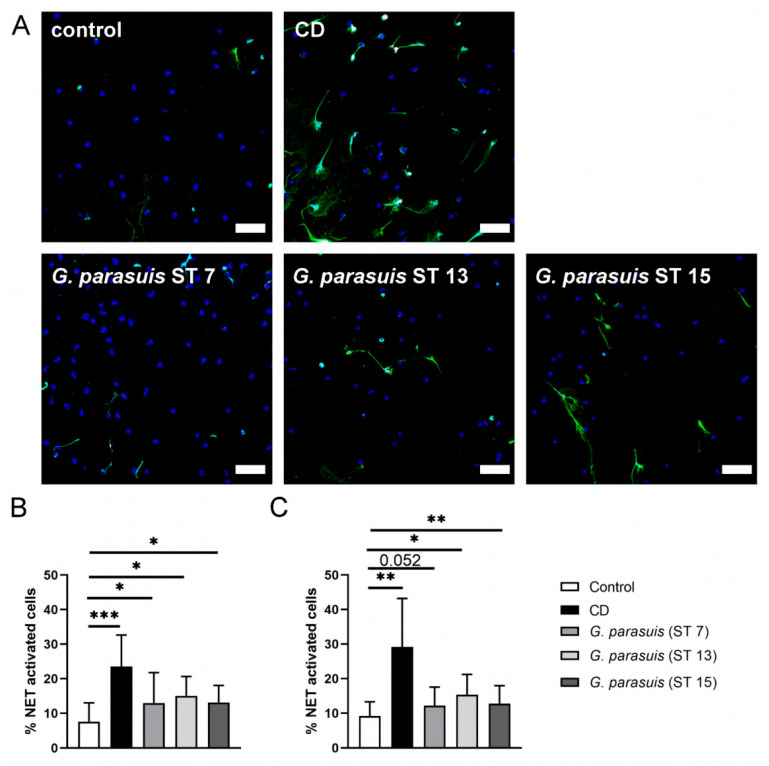
*G. parasuis* induces NETs. (**A**) Representative immunofluorescence images (overlay) of NET induction assays (3 h) that were used for quantification of activated cells are presented. In each experiment and for each sample, six randomly taken pictures from two individual slides were analyzed for quantification. All cells on the six pictures were counted, and the mean of activated cells per stimulus was calculated and used for statistics. RPMI was used as unstimulated control. Methyl-β-cyclodextrin (CD) diluted in RPMI was used as positive control. Staining: Blue = DNA, green = DNA/histone-1-complex, red = myeloperoxidase, scale bar = 50 µm. All settings were adjusted to a respective isotype control. (**B**) Statistical analysis of NET induction assay after 1 h stimulation. The three serotypes from *G. parasuis* significantly stimulated neutrophils to release NETs. (**C**) Statistical analysis of NET induction assay after 3 h stimulation. The three serotypes of *G. parasuis* induced in a comparable amount NETs as already after one hour of infection. Data are presented with mean ± SD and were analyzed with one-tailed paired Student’s *t*-test (*n* = 5) (* *p* < 0.05, ** *p* < 0.01, *** *p* < 0.001).

**Figure 3 pathogens-11-00880-f003:**
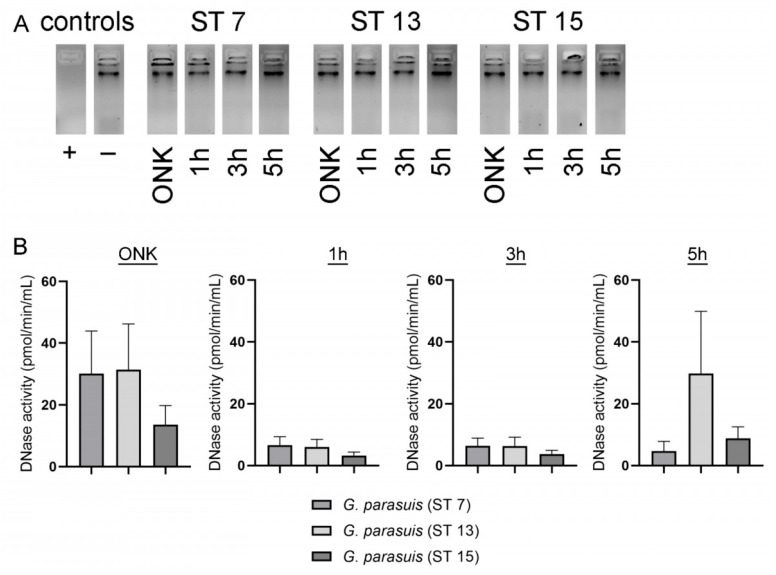
DNase activity was detectable in the supernatant of *G. parasuis* ST 7, 13 and 15 with a sensitive DNase activity assay. (**A**) The supernatants from different growth phases of *G. parasuis* serotypes were firstly analyzed for their degradation capacity of eukaryotic DNA. After 24 h incubation of calf thymus DNA in presence of a DNase buffer (final concentration 1.5 mM MgCl_2_ and 1.5 MM CaCl_2_ and pH 7.4) and *G. parasuis* supernatants, no degradation was detectable in all samples compared to a positive control (+). The samples were analyzed by 1% agarose gel electrophoresis. (**B**) By analysis with a more sensitive DNase activity assay, a growth phase-dependent increase in DNase activity was detectable in all serotypes, and the highest value was detected in the overnight culture (ONK). The activity measurement is presented from three independent experiments, and data are presented with mean ± SEM.

**Figure 4 pathogens-11-00880-f004:**
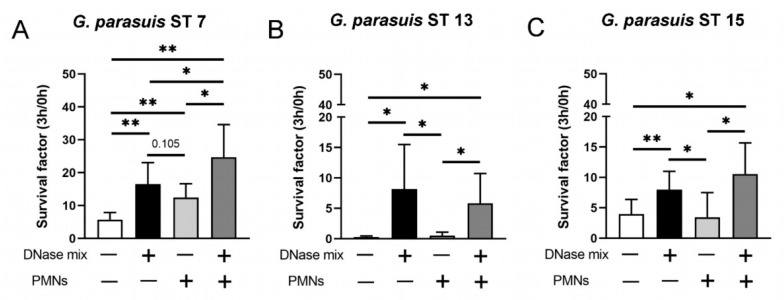
*G. parasuis* ST 7 survives best in presence of degraded NETs. (**A**–**C**) *G. parasuis* serotypes were grown for 3 h at 37 °C in the absence or presence of neutrophils (PMN) and a DNase mix (DNase I and micrococcal nuclease). *G. parasuis* ST 7 survives best in presence of degraded NETs. The survival factor was calculated (*n* = 5). All data are presented with mean ± SD and were analyzed with one-tailed paired Student’s *t*-test (* *p* < 0.05, ** *p* < 0.01).

**Figure 5 pathogens-11-00880-f005:**
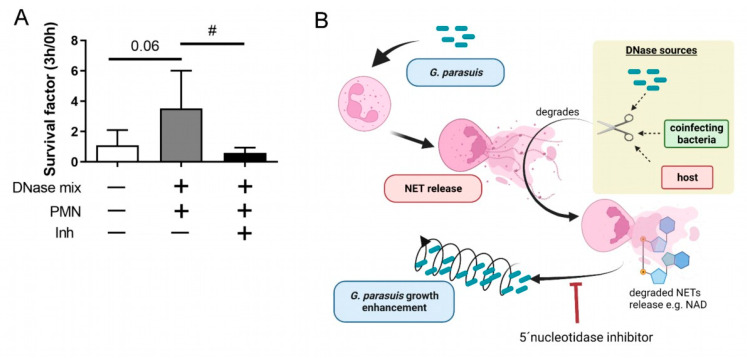
(**A**) *G. parasuis* ST 15 growth in presence of degraded NETs is inhibited by 5′-nucleotidase inhibitor. Freshly grown and washed *G. parasuis* ST 15 was grown for 3 h at 37 °C in the presence of neutrophils (PMN) and a DNase mix (DNase I and micrococcal nuclease). As inhibitor (Inh) 500 µM 5′-nucleotidase inhibitor was added to reduce adenosine and NAD metabolism. The survival factor was calculated (*n* = 4). All data are presented with mean ± SD. Data were analyzed with one-tailed paired Student’s *t*-test to the negative control and between DNase mix + PMN +/− Inh (# *p* < 0.05). (**B**) Hypothetical model based on the findings in this study (prepared with BioRender.com).

**Table 1 pathogens-11-00880-t001:** BLAST results of bacterial DNases in *G. parasuis*. The table presents an overview of described bacterial DNases that were not identical or partially identical found in *G. parasuis*.

Bacteria	Dnase Name	References
*Vibrio cholerae*	Xds and Dns	[28,29]
*Neisseria gonorrhoeae*	Nuc	[30]
*Staphylococcus aureus*	Nuc	[31]
*Streptococcus suis*	SsnA	[27,32,33]
*Streptococcus pneumoniae*	EndA	[32,33,34,35,36,37]
*Streptococcus pyogenes*	Sda1 and MF	[38,39,40,41,42,43]

## Data Availability

The authors confirm that the data supporting the findings of this study are available within the published article. Raw data were generated at the Department of Biochemistry, University of Veterinary Medicine Hannover, Foundation, Hannover, Germany. Derived data supporting the findings of this study are available from the corresponding author, N.d.B, after request.

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
