# Peer review of "Studying the Interaction of Neutrophils and Glaesserella Parasuis Indicates a Serotype Independent Benefit from Degradation of NETs"

_pathogens, 2022, doi:10.3390/pathogens11080880_

Round 1

Reviewer 1 Report

Bonilla and co-workers studied a topic very interesting and important to swine industry. In this study, they investigated the interaction between neutrophils and  three different resotypes of Glaesserella parasuis.

The study is very inteteresting; however, before pubblication need of some revision.

1-Introduction: the authors should  clarify why they choose serotypes 7, 13 and 15 for your study.

2-Lines 87-97 this part should be reduced or delete in the introduction and insert in discussion

3-Results and M&M are clear

4-Discussion, line 299-306. The authors should  better  underline the importance of their study. The conclusions are not clair and the paper loses its originality

Author Response

Answers to the reviewers Pathogens- 1818678 - first report

Dear Editors and Reviewers,

We thank the reviewer’s for the first report and the constructive suggestions.

Please find below our answers to the comments and questions of the reviewers.

We have prepared a revised version of the manuscript, highlighting the changes from first revision using the Track Changes function in Microsoft Word, and we have prepared a point-to-point response in order to address the remarks out by the reviewers.

We thank again the reviewers for the comments to our study and tried to improve the manuscript based on the constructive comments.

With kind regards

Nicole de Buhr

Reviewer 1

Comments and Suggestions for Authors

Bonilla and co-workers studied a topic very interesting and important to swine industry. In this study, they investigated the interaction between neutrophils and  three different resotypes of Glaesserella parasuis.

The study is very inteteresting; however, before pubblication need of some revision.

  1. Introduction: the authors should clarify why they choose serotypes 7, 13 and 15 for your study.

Answer: We thank the reviewer for this comment and extended this section stating, why we included these serotypes in line 110-114.

“Firstly, these three strains were chosen to compare isolates from different origin organ. Secondly, these serotypes cover the range of virulence known for G. parasuis from avirulent to highly virulent. With serotype 7 and 13 we included two of the most common serotypes in North America, Europe and Asia [25].”

  1. Lines 87-97 this part should be reduced or delete in the introduction and insert in discussion

Answer: We thank the reviewer for this recommendation and have shorten the lines 86-91

“For Actinobacillus pleuropneumoniae (A.pp), which is also a member of the Pasteurellaceae, it was shown by us that degraded NETs cause a growth enhancement [20]. As degraded NETs release NAD and adenosine, which promotes the growth of A.pp. This phenotype could be abolished by inhibiting the 5’-nucleotidase. This is described as an essential enzyme in the NAD and adenosine metabolism of cells and bacteria [21]. A.pp as a NAD dependent growing bacteria benefits from the host defense mechanism.”

  1. Results and M&M are clear

Answer: We thank the reviewer for the proof-reading of the section Material and Methods.

  1. Discussion, line 299-306. The authors should  better  underline the importance of their study. The conclusions are not clair and the paper loses its originality

Answer: We thank the reviewer for this comment and reworked the conclusion as following:

“In conclusion, we identified that serotypes 7, 13 and 15 of G. parasuis induce only slightly ROS production, which indicates that the observed NET induction is ROS-independent. NET activated cells can be observed although all G. parasuis serotypes release a DNase. Therefore, G. parasuis can escape from NETs and all investigated serotypes benefit in their growth from degraded NETs. As G. parasuis produces a DNase, another external DNase source, as described for A.pp, this is not obligatory needed for an enhanced growth. However, the external DNase is supporting the growth of G. parasuis. If a pig is infected, other host-defense mechanisms of neutrophils than NET formation are needed to counteract a severe infection, as for example an antibody based phagocytosis. The interaction of neutrophils and G. parasuis serotypes show partially differences and varying degrees of intensity that should be investigated in future studies more in detail. These differences may explain why some strains can lead to meningitis and others cannot. ”

Reviewer 2 Report

Please improve summary: ideas expressed on lines 30 to 35 are not clear and might lead to erroneous interpretation by the reader. For instance: “Interestingly, we detected that G. parasuis induces serotype independent NET activated cells to a small but significant extent”. It induces NET activated cells, OR it induces NET formation of cells independently of the serovars?  As it is is not clear and do are the other sentences.

M&M

How many pigs were used as neutrophils donors?

Were all G. parasuis SV tested with neutrophils from the same pig?

Pigs were previously tested to assure they were free from any infection that could trigger neutrophils in vivo?

Author Response

Answers to the reviewers Pathogens- 1818678 - first report

Dear Editors and Reviewers,

We thank the reviewer’s for the first report and the constructive suggestions.

Please find below our answers to the comments and questions of the reviewers.

We have prepared a revised version of the manuscript, highlighting the changes from first revision using the Track Changes function in Microsoft Word, and we have prepared a point-to-point response in order to address the remarks out by the reviewers.

We thank again the reviewers for the comments to our study and tried to improve the manuscript based on the constructive comments.

With kind regards

Nicole de Buhr

Comments and Suggestions for Authors

  1. Please improve summary: ideas expressed on lines 30 to 35 are not clear and might lead to erroneous interpretation by the reader. For instance: “Interestingly, we detected that  parasuis induces serotype independent NET activated cells to a small but significant extent”. It induces NET activated cells, OR it induces NET formation of cells independently of the serovars?  As it is is not clear and do are the other sentences.

Answer: We thank the reviewer for that comment and have rewritten the summary to avoid misunderstandings (lines 30-33):

“Interestingly, we detected that independent of the serotype of G. parasuis NET formation in neutrophils was to a small but significant extent induced. This phenomenon occurred despite the ability of G. parasuis to release nucleases, which degrade NETs. Furthermore, the growth of Glaesserella enhanced by external DNases and degraded NETs.”

  1. M&M: How many pigs were used as neutrophils donors?

Answer:

In total 12 pigs were included as a blood donor in this study.”

  1. Were all G. parasuis SV tested with neutrophils from the same pig?

Answer:

“Within the assays that show data with the three serotypes, always neutrophils from the same donor pig (for one independent experiment) were used for all serotypes in parallel.”

  1. Pigs were previously tested to assure they were free from any infection that could trigger neutrophils in vivo?

Answer:

The donor pigs were kept either in the Clinic for Swine, Small Ruminants and Forensic Medicine and Ambulatory Service at the University of Veterinary Medicine Hannover, Germany for sperm donation or in the Research Center for Emerging Infections and Zoonoses from the University of Veterinary Medicine Hannover, Germany as control animals. Pigs included in this study were male and female, were not under treatment and a general health examination by a trained veterinarian was conducted before the blood donation.

Reviewer 3 Report

Bonilla et al. evaluated the NETs formation induced by  G. parasuis. The topic is interesting and in the aims of this journal. The manuscript is well written and, in general, clear. However, I found several weaknesses that reduce the quality of this study. 

Results section: 

2.1

I can't entirely agree with the authors; there is not a "slight" increase. There are no differences. In  Figure 1,  I suggest showing MFI in Figure 1C because Fig. 1 D shows this information. In my opinion, the authors must use the untreated stained as a control, instead untreated unstained cells. 

2.2.

In my opinion, the significant difference reported in Figures 2B and 2C is not accurate. Bars look similar to the control, and SD is high. Positive control looks good. 

3.3.

Similar problem. The results description is too speculative. Authors claim the "strongest increase in serotype 13", but the SD is high, and no significant differences are shown.

2.4 

Because there is no evidence of NETs, the assumption that NETs killed G. parasuis lacks support. The survival factor has to be explained by other causes. 

Discussion is speculative and supported by results that lack significance. For example, in Lines 235-236: The authors claim that G.parasuis is significantly inducing NETs compared to the medium control (Fig 2.) Fig 2 does not show significant differences. 

In my opinion, this study did not have the merits to be considered in Animals. 

Minor:

Lines 105-106. Call the attention that serotype 15 was isolated from an "unknown organ". Please confirm 

Round 2

Reviewer 3 Report

This Reviewer appreciates the comments and response to my suggestions and recommendations. 

1. The authors have modified the figure legend, but the analysis and final results are the same. The new legend: "The untreated and stained cells were used as a control." This statement is not valid. Figure 1D shows the MFI untreated and unstained cells, as in the previous version. 

2. Thank you, the authors, for providing the data and the analysis. In the case of Figure 2B, the mean of ST7, ST13, and ST15 is 15.06, 12.98, and 13.13. ST7 showed a higher value. But figure 2D shows ST13 with a higher mean (the bar is higher than ST7 and ST15). Same problem with Figure 2C. The data provided by the authors did not match the figures' results. I suppose that there is a mistake.

3. Regarding figure 3, where the authors only show the results of two experiments, I will let the final decision to the Editor in chief if he agrees that these data support conclusions. In my opinion, at least 3 replicates are needed. 

4. Ok

5. Line 240: "....significantly inducing NETs compared to the medium control." In my opinion, Fig 2 did not show a "significant" increase compared to the medium control. The statement is overestimated.

Lines 243-247. Please confirm that the discussion refers to Fig 1B and 1C. It seems that the authors refer to Fig 2B and 2C.

Lines 303-314. The authors support this conclusion with two independent experiments and data with a high SD. 

Round 3

Reviewer 3 Report

I have no additional comments.